# Application of CRISPR/Cas9 Technology in Cancer Treatment: A Future Direction

Ali A. Rabaan [1,2,3,*], Hajir AlSaihati [4], Rehab Bukhamsin [5], Muhammed A. Bakhrebah [6], Majed S. Nassar [6], Abdulmonem A. Alsaleh [7], Yousef N. Alhashem [7], Ammar Y. Bukhamseen [8], Khalil Al-Ruhimy [9], Mohammed Alotaibi [9], Roua A. Alsubki [10], Hejji E. Alahmed [11], Saleh Al-Abdulhadi [12,13], Fatemah A. Alhashem [14], Ahlam A. Alqatari [15], Ahmed Alsayyah [16], Ramadan Abdelmoez Farahat [17], Rwaa H. Abdulal [18,19], Ali H. Al-Ahmed [20], Mohd. Imran [21] and Ranjan K. Mohapatra [22,*]

1 Molecular Diagnostic Laboratory, Johns Hopkins Aramco Healthcare, Dhahran 31311, Saudi Arabia
2 College of Medicine, Alfaisal University, Riyadh 11533, Saudi Arabia
3 Department of Public Health and Nutrition, The University of Haripur, Haripur 22610, Pakistan
4 Department of Clinical Laboratory Sciences, College of Applied Medical Sciences, University of Hafr Al Batin, Hafr Al Batin 39831, Saudi Arabia
5 Dammam Regional Laboratory and Blood Bank, Dammam 31411, Saudi Arabia
6 Life Science and Environment Research Institute, King Abdulaziz City for Science and Technology (KACST), Riyadh 11442, Saudi Arabia
7 Clinical Laboratory Science Department, Mohammed Al-Mana College for Medical Sciences, Dammam 34222, Saudi Arabia
8 Department of Internal Medicine, College of Medicine, Imam Abdulrahman Bin Faisal University, Dammam 34212, Saudi Arabia
9 Department of Public Health, Ministry of Health, Riyadh 14235, Saudi Arabia
10 Department of Clinical Laboratory Sciences, College of Applied Medical Sciences, King Saud University, Riyadh 11362, Saudi Arabia
11 Department of Laboratory and Blood Bank, King Fahad Hospital, Al Hofuf 36441, Saudi Arabia
12 Department of Medical Laboratory Sciences, College of Applied Medical Sciences, Prince Sattam Bin Abdulaziz University, Riyadh 11942, Saudi Arabia
13 Saleh Office for Medical Genetic and Genetic Counseling Services, The House of Expertise, Prince Sattam Bin Abdulaziz University, Dammam 32411, Saudi Arabia
14 Laboratory Medicine Department, Hematopathology Division, King Fahad Hospital of the University, Al-Khobar 31441, Saudi Arabia
15 Hematopathology Department, Clinical Pathology, Al-Dorr Specialist Medical Center, Qatif 31911, Saudi Arabia
16 Department of Pathology, College of Medicine, Imam Abdulrahman Bin Faisal University, Dammam 31441, Saudi Arabia
17 Faculty of Medicine, Kafrelsheikh University, Kafr El-Shaikh 33511, Egypt
18 Department of Biology, Faculty of Science, King Abdulaziz University, Jeddah 21589, Saudi Arabia
19 Vaccines and Immunotherapy Unit, King Fahad Medical Research Center, King Abdulaziz University, Jeddah 21589, Saudi Arabia
20 Dammam Health Network, Eastern Health Cluster, Dammam 31444, Saudi Arabia
21 Department of Pharmaceutical Chemistry, Faculty of Pharmacy, Northern Border University, Rafha 91911, Saudi Arabia
22 Department of Chemistry, Government College of Engineering, Keonjhar 758002, India
* Correspondence: arabaan@gmail.com (A.A.R.); ranjank_mohapatra@yahoo.com (R.K.M.)

**Abstract:** Gene editing, especially with clustered regularly interspaced short palindromic repeats associated protein 9 (CRISPR-Cas9), has advanced gene function science. Gene editing's rapid advancement has increased its medical/clinical value. Due to its great specificity and efficiency, CRISPR/Cas9 can accurately and swiftly screen the whole genome. This simplifies disease-specific gene therapy. To study tumor origins, development, and metastasis, CRISPR/Cas9 can change genomes. In recent years, tumor treatment research has increasingly employed this method. CRISPR/Cas9 can treat cancer by removing genes or correcting mutations. Numerous preliminary tumor treatment studies have been conducted in relevant fields. CRISPR/Cas9 may treat gene-level tumors. CRISPR/Cas9-based personalized and targeted medicines may shape tumor treatment. This review examines

CRISPR/Cas9 for tumor therapy research, which will be helpful in providing references for future studies on the pathogenesis of malignancy and its treatment.

**Keywords:** CRISPR/Cas; advanced technologies; advanced therapeutics; cancer; cancer treatment; oncoviruses; clinical trials

## 1. Introduction

Clustered regularly interspaced short palindromic repeats associated protein 9 (CRISPR-Cas9) is the natural immune defense system (acquired) mediated by RNA that is present in archaea and bacteria [1,2]. As its name indicates, single-guide RNAs (Cr and Cas9 nuclease) are its major constituents [3]. It encodes a guide RNA, and double-stranded breaks (DSBs) are produced at a particular location of DNA that is being targeted. The direct binding of Cas9 nuclease and a target DNA sequence occurs, which generates DSBs [4]. CRISPR/Cas9 is a facilitator compared to transcription-activator-like effector nucleases (TALENs) [5] and zinc-finger nucleases (ZFNs) [6], as it can target multiple DNA sites at a time through multiplexing. For the joining of DSBs, two different mechanisms are present, including non-homologous end joining (NHEJ) [7] for non-homologous sequences and homologous directed repair (HDR) [8] for homologous sequences [9].

There are some other tools for editing the genome, including ZFNs [10,11] and TALENs [12,13], in which DNA-binding domains of transcription factors and the nuclease domain of the restriction enzyme FokI fuse together. Nuclease activity is activated when FokI domain site-specific nucleases form dimers when paired adjacent sequences are targeted, forming DSBs near the binding sites. The fusion of DNA-binding proteins such as TALENs, ZFNs, and dead CAS system 9 (dCas9) with fluorescent proteins (FPs), along with their role in genome editing, also helps in the direct imaging of genomic loci in living cells [14,15] (Table 1).

**Table 1.** Comparison and working apparatus of genome engineering tools.

| Properties | CRISPR | ZNFs | TALENs | FLP-FRT | CRE-LOXP | Bibliography |
|---|---|---|---|---|---|---|
| DNA binding moiety | RNA | Protein | Protein | Flippase recombination target | Site-specific recombinases | |
| Ease of targeting multiple targets | High | Low | Low | High | High | |
| Complexity of design | Simple | Very complex | Complex | Simple | Simple | [16–21] |
| Nuclease | Cas | FokI | FokI | Recombinase | Recombinase | |
| Off-target effects | Variable | Moderate | - | Specific | Specific | |
| Toxicity | Low | Variable to high | Low | Low | Variable | |
| Target recognition size | 22 nucleotides | 18–36 nucleotides | 30–40 nucleotides | 20–35 nucleotides | 38 nucleotides | |

Globally, the third largest cause of mortality is cancer, causing 20% of deaths in Europe [22,23]. It is a step-wise progressive disease that results in the halting of growth suppressors [24] and cell death signals [25] due to epigenetic changes in the cellular genome and the buildup of mutations, which also promote an increase in genetic instability during tumorigenesis [26]. Proinflammatory activity, angiogenesis, evasion of the immune system, and invasiveness are the prominent features responsible for the progression of cancer [27]. Driver mutations are responsible for deactivating suppressor genes or the activation of oncogenes [28], whereas passenger mutations are responsible for alterations of the genes in

cancer cells [29]. The use of CRISPR/Cas9 technology in cancer immunotherapy could be a revolutionary approach that promises to increase the chances of recovery among cancer patients [30]. In order to treat cancer, this technique is very helpful in the manipulation of the cancer genome [31], cancer immunotherapy [32], and the inactivation or elimination of viral infections promoting cancer and the epigenome [33].

### 1.1. Behind CRISPR-Cas9

In 1987, a short palindromic repeat sequence was revealed, which was given the name "CRISPR" in 2002 [34]. In 2012, mature CRISPR/RNAs (crRNAs) and transactivating CRISPR RNA (tracrRNA) produced a particular double-stranded RNA structure using complementary base pairing, instructing the Cas9 protein to produce breaks in double strands of the target DNA [35]. The type II Cas system was used to cut DNA in mammalian cells in 2013, which made it possible to use the CRISPR/Cas9 system for the editing of genes [36]. The CRISPR/Cas9 technology has developed very fast, and in 2020 there was a wide variety of available tools that were based on CRISPR/Cas9 for the editing of genes at the RNA and DNA levels [37] (Table 2).

**Table 2.** The cascade contribution in the CRISPR system.

| Years | Findings | Bibliography |
|---|---|---|
| 1987 | Discovery of the CRISPR clustered repeats | [38] |
| 2000 | Acceptance of the widespread presence of CRISPR families in prokaryotes | [39] |
| 2002 | The Cas gene was discovered and given the name "CRISPR." | [34] |
| 2005 | Adaptive immunity function was proposed, and foreign origins of spacers were identified using PAM | [40] |
| 2007 | First experimental proof that CRISPR conferred adaptive immunity | [41] |
| 2008 | CRISPR acts upon DNA target<br>Discovered the function of crRNA | [42,43] |
| 2009 | Cleavage of RNA by Type III B Cmr CRISPR complex | [44] |
| 2010 | Cleavage of target DNA via DSBs through Cas9 was guided by spacer sequences | [45] |
| 2011 | Discovery of tracrRNA in conjunction with Cas9 that formed a duplex structure with crRNA | [45] |
| 2012 | Characterization of Cas9's DNA targeting in vitro | [46] |
| 2013 | Mammalian cell genome editing for the first time<br>Discovery of dCas9, CRISPRi, and CRISPRa | [47] |
| 2014 | Crystal structure of Cas9 in guide RNA and target DNA, genome-wide functional screening with Cas9, and crystal structure of apo-cas9 | [48–50] |
| 2015 | CRISPR/Cas9 was used to edit human embryos but with prominent off-target effects, CRISPR/Cas9 was used to develop virus-resistant tomato plants, and discovery of Cas 12a (Cpf1) | [51,52] |
| 2016 | The invention of base editor (BE)<br>Discovery of Cas13a (C2c2) | [53] |
| 2019 | The invention of nCATS by CRISPR/Cas9 | [54] |
| 2020 | Discovery of the vfCRISPR | [55] |

PAM: protospacer adjacent motif. crRNA: CRISPR/RNAs. tracrRNA: transactivating CRISPR RNA. Cmr: CRISPR RAMP module. dCAS: dead CAS system 9. CRISPRi: CRISPR interference. CRISPRa: CRISPR activation.

### 1.2. CRISPR/Cas9 Apparatus

Bacteria and archaea have adopted an immune system guided by RNA, which is encoded with CRISPR loci and CRISPR-associated genes, which provide immunity (adaptive) against infections of bacteriophages and the transfer of plasmids [56]. Short pieces of foreign DNA are inserted into the host chromosome's CRISPR repeat spacer as new spacers following exposure to invading genetic elements from plasmids or phages during

the process of immunization [57]. As a result of this, the host cell saves this memory for future protection from the same invader [58] (Figure 1).

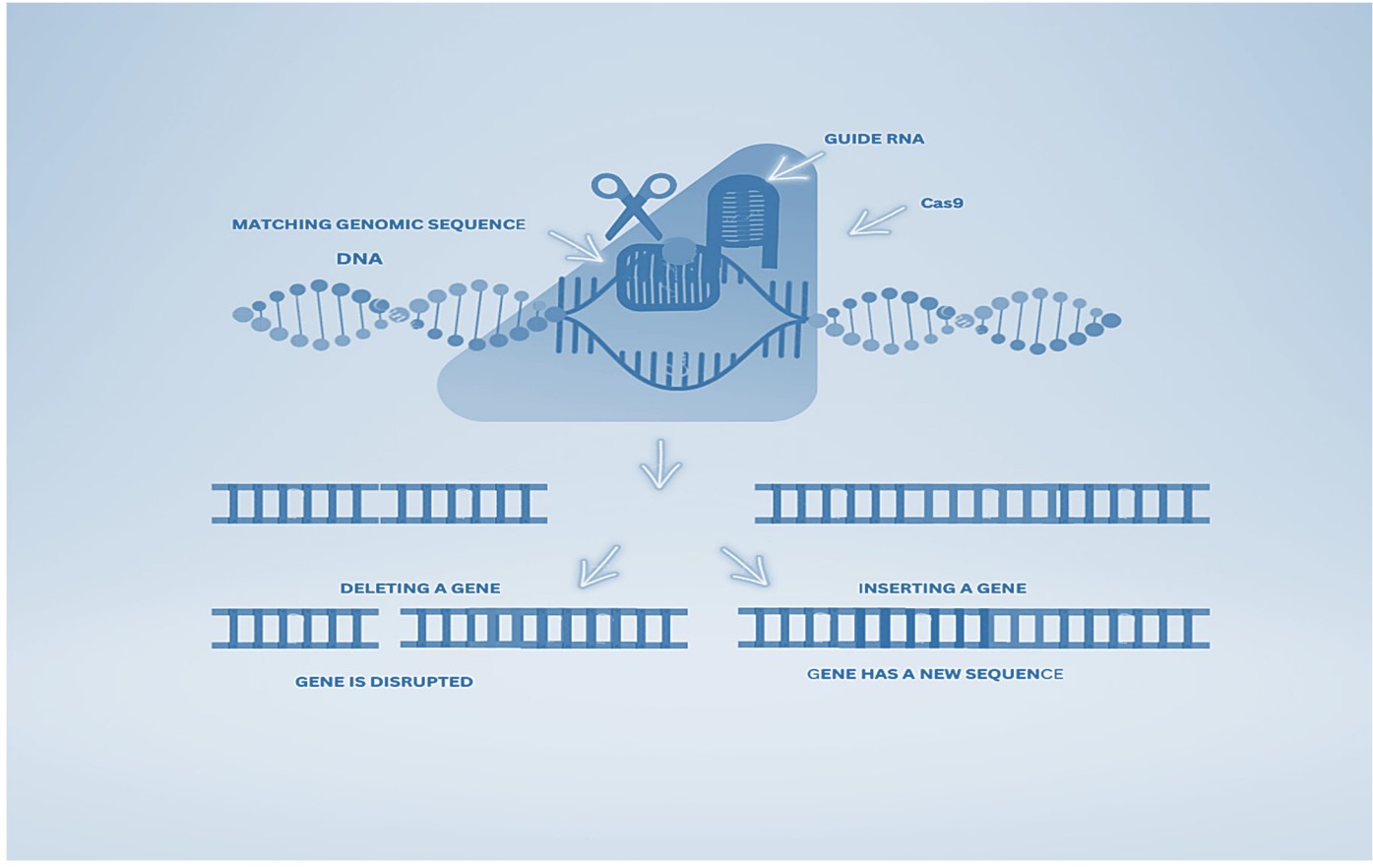

**Figure 1.** Operational scheme of CRISPR-Cas9.

Endonucleolytic cleavage produces mature and short crRNAs due to the CRISPR array's transcription and the enzyme processing of CRISPR precursor transcripts [59]. At the 5′ end, the sequence is complementary to the foreign genetic element, known as a spacer, which is a short fragment of RNA, whereas at the 3′ end, the repeat sequence is present [60]. The Cas nucleases destroy the RNA or DNA due to the hybridization of the complementary target sequence of the foreign genome and the crRNA spacer, also called a protospacer, during the second infection. [61]. One of the crucial aspects of the CRISPR-Cas system is the formation of crRNA–effector complexes due to the assembly of Cas proteins and crRNAs for the integration of DNA targets and the destruction of similar sequences in foreign nucleic acids [62]. In the majority of CRISPR-Cas systems, a protospacer adjacent motif (PAM) is a short conserved sequence of about (2–5 bp) that is present near the crRNA-targeted sequence and has a promising role in the selection of the DNA target and degradation [63].

## 2. The Era of CRISPR/Cas9 in Therapeutic Oncology

CRISPR/Cas9 is considered a source of treatment for cancer. It is being used globally to treat different types of cancers such as brain cancer, renal cell carcinoma, colorectal cancer, hepatocellular carcinoma, urinary bladder cancer, etc.

### 2.1. CRISPR/Cas9 in Brain Cancer

Irrespective of the patient's gender or age, brain cancer has the highest mortality rate of all cancers [64]. For the last five decades, the same therapies have been used against brain cancers, including gliomas [65]. Researchers are facing technical difficulties at the

genetic level to cope with this lethal disease, but CRISPR/Cas9 is a quick and effective technique [66]. In a study conducted on the medulloblastomas and gliomas of human brain cancer, four different types of animal models were used, including cell-derived xenografts (CDXs) [67], an in vivo mouse model, patient-derived xenografts (PDXs) [68], and genetically engineered mise. Through the CRISPR/Cas9 technique, the Nf1, Pten, and Trp53 genes responsible for glioblastoma and the Ptch1 gene accountable for medulloblastomas were knocked out. By using this technique, genes involved in brain tumors can also be knocked out [69].

Besides gene manipulation in the organism's lineage, presumptive tumorigenesis or tumor suppressor genes have long been recognized as the benchmark for simulating carcinoma, especially brain tumors, in mice [70]. According to reports, gain-of-function polymorphisms in a proto-oncogene and loss-of-function alterations in tumor suppressor genes are the two main causes of glioblastoma. It has been determined that the carcinogenic alteration of stem cells from embryos and the loss of function of tumor suppressor genes are caused by homologous recombination-based classical cell-type-specific knockdown approaches [71]. However, due to the lengthy process of genetically engineered murine model (GEMM) creation and the ineffectiveness of gene duplication, its uses are restricted. In contrast, the double-stranded break sites in target genomes are modified more successfully and accurately using the CRISPR/Cas9-guided endonuclease technique. One of the key benefits of CRISPR/Cas9 is the speed at which a GEMM model may be produced. Because it is more capable of understanding pathological conditions than conventional genetic manipulation models, this flexible method of genome engineering has been utilized to develop gene knockout models of both mice and rats, among other animals [72].

### 2.2. CRISPR/Cas9 in Hepatocellular Carcinoma

Different tumor-suppressing genes in the liver have been targeted with CRISPR-Cas9 in a variety of ways in hepatocellular carcinoma [73]. In one approach, through hydrodynamic tail vein injections, the tumor suppressor genes p53 and Pten, either separately or together, were targeted [74]. Liver tumors resembling those in CRE-loxP-deleted Pten and p53 transgenic animals can be formed if p53 and Pten sgRNAs are used in combination. Hepatocellular carcinoma has been successfully treated with CRISPR-Cas9 by navigating the extended process of genetically modified strains with cyclization recombinase locus of crossing over, x, P1 (CREloxP) technology [75].

Hepatocytes have demonstrated that CRISPR/Cas9 can fix a Fah mutation in a mouse model of hereditary tyrosinemia type I. They co-injected single-stranded DNA (ssDNA) with Cas9, sgRNA, the wild-type G nucleotide, and homology arms flanking the sgRNA target area into the mouse model using a non-viral hydrodynamic injection. Less than 1/250 cells underwent early genetic repair in this experiment. Developing Fah-positive hepatocytes can reverse weight loss in a mouse model of hereditary tyrosinemia type I. Even so, only 0.4% of hepatocytes underwent hydrodynamic injection correction. When that happened, a safer and more effective method of CRISPR delivery was considered [76].

The hereditary tyrosinemia type I mouse model has been effectively treated with a cutting-edge therapy termed metabolic pathway remodeling. The second stage of tyrosine catabolism is started by the enzyme hydroxyphenylpyruvate dioxygenase. Using an in vivo CRISPR/Cas9 deletion of hydroxyphenylpyruvate dioxygenase, hepatocytes have been changed from benign tyrosinemia type III to tyrosinemia type I. After that, the entire liver was quickly replaced by modified hepatocytes [77]. By removing the hydroxyphenylpyruvate dioxygenase, tyrosine catabolism can be changed, limiting the buildup of harmful catabolites and tyrosine. In contrast to gene therapy, metabolic pathway reprogramming does not necessitate the ongoing expression of the disease-causing gene's wild-type protein, which may trigger an immunological response, restricting its long-term expression [78].

### 2.3. CRISPR/Cas9 in Colorectal Cancer

This type of cancer targets the rectum or colon. Different genes are targeted and mutated in this type of cancer, as revealed by tumor sequencing studies. Genes that are mutated are involved in tumor progression, tumor phenotype, and carcinogenesis [79]. By using mouse models that were genetically modified, this technique was found to be helpful in the orthotropic organoid transplantation of mice to correct the Trp53 and APC tumor suppressor genes in colon epithelial cells. It can be further used in different treatments for determining the types of mutations occurring in transforming cells that promote growth advantages in multiclonal tumors [80].

With the aid of several high-throughput genomic sequence identification methods, critical genes that contribute to medication resistance in human malignancies have been objectively identified. Many individuals previously employed RNA interference (RNAi) profiling with an shRNA repository to silence specific genes [81]. However, their usage was limited by wasteful quality-lowering and off-target effects. The CRISPR-Cas9 library system, which combines improvements in genome editing technology, has lately presented an alternate strategy to overcome these restrictions. It has been used to pinpoint the genes that are essential for the growth, viability, and medication resistance of cancer cells both in vitro and in vivo [82].

### 2.4. CRISPR/Cas9 in Renal Cell Carcinoma

The tubular cells of the kidney are prone to renal cell carcinomas (RCC) and a tumor type known as clear-cell RCC (ccRCC) [83]. Five types of miRNA, including miR-1274, miR-224, miR-1290, miR-210-3p, and miR-885-5p, are known to be upregulated in ccRCC [84,85], miR-1274a and miR-1251 5p [86,87]. CRIPSR-Cas9 is known to be effective in a metastatic renal cell carcinoma (mRCC) disease in which the tumor suppressor Von Hippel Lindau (VHL) is knocked out. The development of this method allowed for its application to the identification of various RCC-causing genes [88] (Figure 2).

For the diagnosis and regulation of tumor progression and development, the possible biomarker is long non-coding RNA (lncRNA) [89]. Tumorigenesis in cancer of the bladder is related to the upregulation of PANDAR, which is a long non-coding RNA. Different lncRNA genes, including TP53 [90], long non-coding RNA related nuclear protein, and urothelial carcinoma-associated 1 (UCA 1), are associated with carcinoma of the bladder [91]. CRISPR/Cas9 is a technique used for the editing of genes that can be used to manipulate the lncRNA. The transfection of genomic DNA isolated from T24 bladder cancer cells and 5637 cells with CRISPR/Cas9-UCA1 was performed and then observed using DNA sequencing and T7 endonuclease 1 assays [92]. In a study, it was reported that this technique was successful in knocking out the lncRNA-UCA1 [93], promoting the use of this technique in other bladder cancers.

### 2.5. Application of CRISPR/Cas9 in Patient-Derived Organoids

Patient-derived organoids may fill in the gaps left by more conventional culturing techniques in addressing the limitations of cancer stem cells in treatment response prediction [94,95]. Organoids are three-dimensional in vitro cellular structures derived from tissue-specific stem cells, with the ability to self-organize into "mini-organs" resembling the tissue of origin. Organoids provide numerous ways to evaluate therapy responses since they are reasonably easy to maintain and grow, in contrast to other culture methods. The culturing methods may change based on the tissue of origin, much like organotypic tissue slice cultures [96,97]. Using mitogenic stimuli and extracellular matrix, the organoid culture technique promotes the ex vivo growth of tissue-resident stem cells by recreating the microenvironment, or "niche", necessary for stem cell self-renewal. The Cas9 nuclease from Streptococcus pyogenes and the tailored guide RNA used in the archetypal CRISPR-Cas9 system for editing the mammalian genome detect and target a specific DNA sequence that comes before the protospacer neighboring motif sequence [98]. CRISPR-Cas9 permits the creation of a DNA double-strand break at a specified genomic site, despite

the necessity of this motif sequence, which significantly varies across Cas9 variations [99]. Non-homologous end joining (NHEJ) and homology-directed repair are the two processes used to repair double-strand breaks in mammalian DNA (HDR). The biallelic insertion of indel mutations results in gene knockout because the error-prone NHEJ randomly inserts indels throughout the repair process [100].

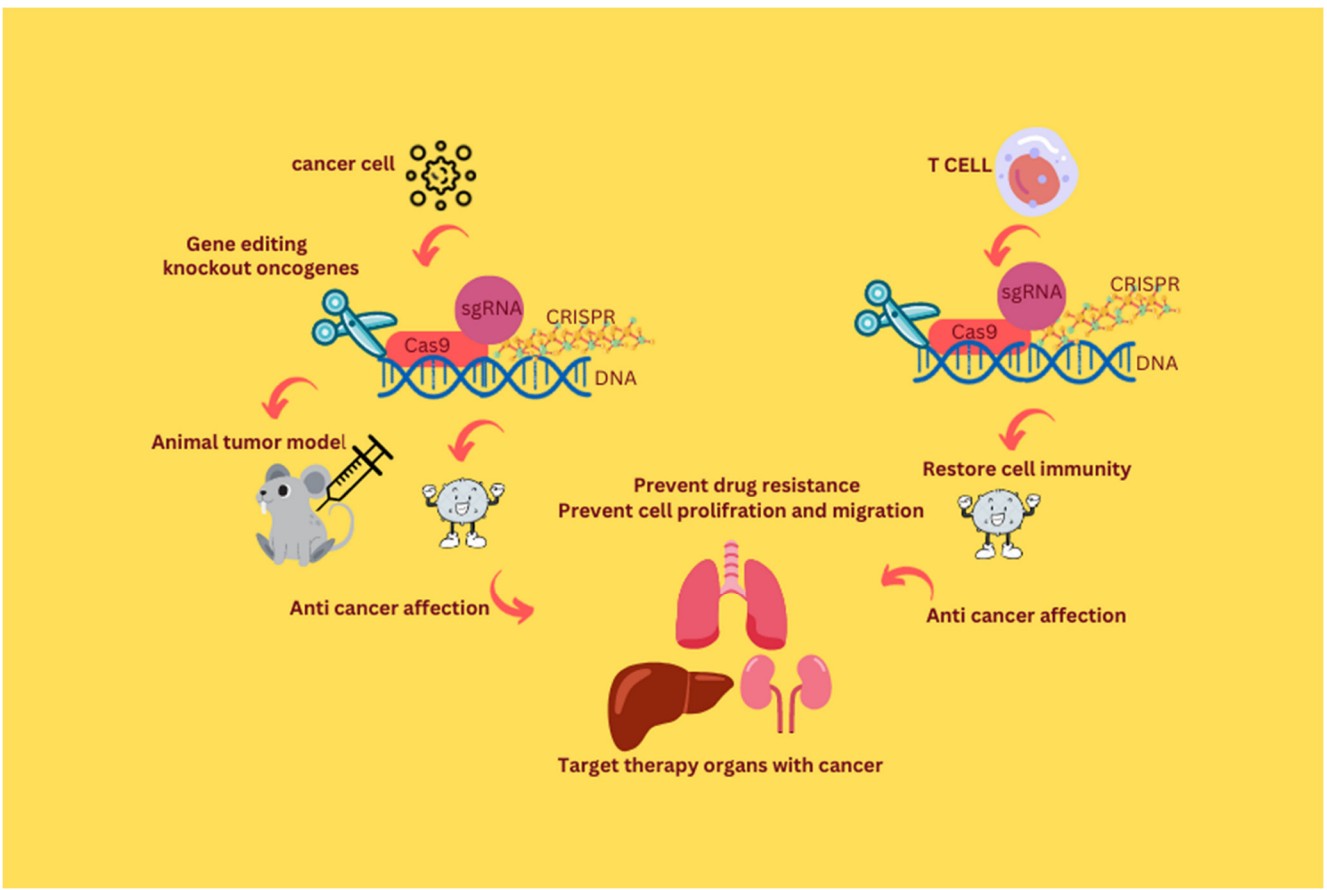

**Figure 2.** CRISPR/Cas genome editing tool in the age of cancer treatment.

When certain DNA templates are given with CRISPR-Cas9, HDR, which is typically used to replace a damaged allele using an intact genome, may be coopted for gene knock-in [101]. For instance, missense mutations are introduced using single-strand oligonucleotides or plasmids with nucleotide variations and homology arms, while selection markers or gene reporters are integrated using HDR templates with functioning gene cassettes [102]. While CRISPR-Cas9 technology has been developed for a variety of applications, including DNA base editing, RNA targeting, gene expression regulation, epigenome editing, and the visualization of particular DNA loci, the use of CRISPR-Cas9 on organoids primarily uses NHEJ and HDR mechanisms to engineer genes of interest. Organoids are in fact ideal instruments for evaluating gene activities through potential genome engineering since the organoid system permits the development of undamaged tissues without sacrificing genetic or phenotypic stability [103].

## 3. CRISPR/Cas9 in Oncolytic Viruses

Oncolytic viruses are the basis of viral vector delivery and can replicate inside cancer cells [104]. Inside cancer cells, oncolytic viruses (OV) benefit from the reduced ability to respond to proapoptotic signals and the resistance developed against apoptosis that shields healthy and active cells from infection [105]. Retroviruses, vaccinia viruses, mumps,

adenoviruses, and herpes simplex virus (HSV) are the viruses most often studied for treatment [106]. These viruses are modified to enhance their efficacy and immunogenicity.

Oncoviruses have undergone different trials to use them as anticancer agents [107]. Their ability to infect neoplasms made them a perfect choice for CRISPR/Cas9 delivery systems. After the failure of traditional viral vectors, to target the NARS gene in xenografts of embryonic rhabdomyosarcoma in mice, the myxoma virus (MYXV) was a suitable vector for CRISPR/Cas9 [108]. It can target multiple sequences due to the vast packaging capacity of OV (161.8 kbp). The survival of mice was increased and the growth of tumors was decreased as a result of the inactivation of NARS mediated by CRISPR/Cas9 [109,110].

## 4. Editing the Cancer Epigenome

Cancers can be treated using the CRISPR/Cas9-mediated epigenome editing tool [111]. In this technique, Cas9 is fused with a transcription activator for activation or with a repressor for repression. Genetic mutations were found to be present in oncogenes or tumor suppressor genes (TSGs) in cancer cells [112]. It is possible for CRISPR/Cas9 to achieve gain-of-function or loss-of-function mutations. It was found to help identify the causative genes in multiple cancers. The relative cancers can be controlled by identifying and targeting these genes [113]. The ER regulator SRC-1 gene has a central role in the progression of metastatic disease by ER tumors. SRC-1 cooperates with ER to regulate a network of cancer-related genes linked to differentiation and proliferation [114]. Due to the silencing of SRC-1 by CRISPR/Cas9-based epigenetics, genes involved in proliferation and differentiation were poorly expressed. As a result, tumor metastasis or breast cancer progression can be controlled effectively [115] (Table 3).

**Table 3.** Targeted cells, malignancy types, vector models, and genome engineering by CRISPR/Cascade 9.

| Disease | Target Cells | Gene/s | Aim/Repair Pathway | Format/Delivery | Reference |
|---|---|---|---|---|---|
| Myeloid malignancies | LSK | (TET2, RUNX1), (SMC3, TET2), (NF1, EZH2, and DNMT3A) | Knock out/NHEJ | Two-vector system/Lentivirus | [116] |
| Myeloid malignancies | RN2 with constitutive Cas9 expression | 192 chromatin regulatory domains | | One-vector system/Lentivirus | [117] |
| MDS | K562 | SRSF2 | Point mutation/HDR | CRISPR vector and ssODN/Electroporation | [118] |
| MDS, CMML, AML | KBM5 | ASXL1 | Mutation correction/HDR | | [119] |
| MLL | HEK293 | MLL and AF4 | Chromosomal rear-rangements/HDR | CRISPR vector and template plasmid/Lipofection | [120] |
| AML | K562 | IDH2 | Knock in/HDR | CRISPR vector and template plasmid/Nucleofection | [121] |
| AML | Primary AML blasts | IDH2 | Mutation correction/HDR | Two-vector system/Lentivirus | [122] |
| SCN | iPSC | HAX1 | | CRISPR vector and ssODN/Lipofectamine | [123] |
| Pediatric AML | Human HPSC | MLL and ENL | Chromosomal rearrange-ments/NHEJ | One-vector system/Lentivirus | [124] |
| AML | Human HPSC | RUNX1 and ETO | | One-vector system/Electroporation | [125] |

**Table 3.** *Cont.*

| Disease | Target Cells | Gene/s | Aim/Repair Pathway | Format/Delivery | Reference |
|---|---|---|---|---|---|
| AML and MDS | Human HPSC | (TET2, U2AF1), (DNMT3A, RUNX1), (ASXL1, TP53), (EZH2, STAG2), (SMC3, TP53, and SRSF2) | Knock out/NHEJ | One-vector system/Lentivirus | [126] |
| MDS | U937 | ASXL1 | | Two-vector system/Electroporation | [119] |
| CHIP | Human HPSC | DNMT3A and TET2 | | One-vector system/Lentivirus | [127] |
| CHIP | LSK | FLT3, DNMT3A, SMC3, EZH2, RUNX1, and NF1 | | RNP/Electroporation | [128] |
| XCGD | PLB | CYBB | Mutation correction/NHEJ | One-vector system/Lentivirus | [129] |

RNA-guided endonuclease gene editing is now carried out using the CRISPR-Cas9 system. An sgRNA and the nuclease Cas9 make up the system's fundamental elements. HNH and RuvC are the two catalytically active domains of the nuclease Cas9. The RuvC domain has three subdomains spread over the linear protein sequence, while the HNH domain is a single nuclease domain. RuvC I is located close to Cas9's N-terminal region, while RuvC II and III, which surround the HNH domain, are located close to the protein's midsection. The complementary and non-complementary strands of the target DNA may be cut by the HNH and RuvC nuclease domains, respectively. The sgRNA is formed from crRNA and tracrRNA, and it contains an invariant scaffold region and a spacer region. Using a 20 nt guide sequence and base pairing to the genomic target, the sgRNA binds to Cas9 and guides it to the region of interest. The Cas9 component of the CRISPR-Cas9 system cleaves DNA 3–4 base pairs upstream of PAM and creates sequence-specific DSBs as a result. The genetic engineering models used in epigenetics using CRISPR/Cas9 technology are shown in Table 4.

**Table 4.** Genetic engineering models used for CRISPR/Cas9 epigenetics.

| Different Approaches | Organisms | Genes | References |
|---|---|---|---|
| Gene knockout | | Invertebrates | |
| | Caenorhabditis elegans | (unc-1, csr-1, dpy-3, and mes-6) | [130,131] |
| | Silkworm | (BmKMO and BmTH), (BmBLOS2 and Bm-ok), and (Bmtan and BmWnt1) | [132–134] |
| | Yeast | (ADE-2) | [135] |
| | Drosophila | Yellow, white, and AGO1 | [136] |
| | | Vertebrates | |
| | Chicken | Stra8 and Myostatin | [137,138] |
| | Human | (MED12 and DMRT1), (OCIAD1 and DMRT3), (NF1 and NF2), (CUL3 and H69), (TADA2B and TADA1), and (MAGEC2 and S100A4) | [139–143] |
| | Mouse | Rp9 | |
| | Zebrafish | cyp19a1a, valopa, and valopb | [144] |
| | Monkey | Ppar-γ and Rag1 | [145] |
| | | Plants | |
| | Rice and Arabidopsis Tobacco Sorghum | (IAA2 and CDK), (PDS3 and OsSWEET11), and (TTG1 and OsSWEET14) | [146–150] |

**Table 4.** *Cont.*

| Different Approaches | Organisms | Genes | References |
|---|---|---|---|
| | | Invertebrates | |
| | Silkworm | Bmku70 | [151] |
| | Drosophila | Yellow locus, white locus, and nanos | [136,152] |
| | Caenorhabditis elegans | unc-119 | [130] |
| | | Plants | |
| | Tobacco | No | [153] |
| Gene knock-in | Arabidopsis | PDS3 and AtFLS2 | |
| | Rice | WDV | [154] |
| | | Vertebrates | |
| | Mouse | Rosa26, KRAS, p53, and LKB1 | [155] |
| | Chicken | yRad52 | [154] |
| | Pig | COL1A | [156] |
| | Human | DACT1, IFIT1, and EGR1 | [157] |
| | Zebrafish | Fus, Zebrafish th, and tardbp | [158] |
| | | Invertebrates | |
| | Caenorhabditis elegant | TRHR-1 | |
| Gene knockdown and silencing approaches | Drosophila | roX1 and roX2 | [159] |
| | Silkworm | No | |
| | | Vertebrates | |
| | Mouse Chicken Pig | No | [160,161] |
| | Zebrafish | EPHA1, mmp21, and Nr1 | |
| | Human | No | |
| | | Invertebrates | |
| | Silkworm | | |
| | Drosophila | No | [162] |
| | Caenorhabditis elegans | | |
| Gene correction | | Vertebrates | |
| | Zebrafish | No | |
| | Chicken | | |
| | Human | MYBPC3 | [163,164] |
| | Pig | No | |
| | Mouse | Hemophilia B and Pde6b | |
| | | Invertebrates | |
| | Caenorhabditis elegans | dpy-5, unc-76, and lon-2 | [165] |
| | Silkworm | No | |
| | Drosophila | wg, bam, cid, nos, ms(3)k81, and wg | [166] |
| Conditional approaches | | Vertebrates | |
| | Zebrafish | tyr, insra; insrb, and ascl1a | [165] |
| | Human | puroR and Ctnnb1 | |
| | Chicken | No | [138] |
| | Pig | PFFs | [167] |
| | Mouse | Kras, Mecp2, Lkb1, Ispd, and p53 | [158,168] |

## 5. Clinical Trials of CRISPR/Cas9

The first clinical trial (ex vivo) on non-small-cell lung cancer patients was performed in China using CRISPR/Cas9 as a tool for the editing of genes [169]. The electroporation of Cas9 and sgRNA was performed, in which the PD-1 gene present in T cells in the peripheral blood of patients was targeted and inculcated back into the patients [170]. In the peripheral blood, edited T cells were found to be present in the patients who received infusions within a very short period. As a result, they discovered that this method was efficient and secure, which improved therapeutic efficacy [33].

A phase 1 in-human CRISPR-Cas9-technology-based clinical trial was conducted in which three patients with refractory cancer at an advanced stage were enrolled, which was recently reported by White et al. [171]. Genes that were encoding chains of endogenous PDCD1 and TCR, i.e., TRBC and TRAC, were taken out of the T lymphocytes of the patients

to boost antitumor immunity. The introduction of a transgene (NY-ESO-1) allowed for the detection of tumors. The patients were able to tolerate the engineered T lymphocytes for up to 9 months after they were reintroduced [172].

Another clinical trial was carried out for CD19 tumor cells and suggested CAR-T-cell therapy for relapsed hematological malignancies. At the TRAC locus of cells that can effectively recognize CD19 cells, the integration of CARs, including CD20 or CD22 and CD19, was carried out [173]. To deliver CARs via a lentivirus (LV) to patients with relapsed or resistant CD19+ lymphoma and leukemia, gene-disrupted allogeneic universal CD19-specific CAR-T cells (UCART019) were used in a different trial [174]. Endogenous genes, such as B2M and TCR, were disrupted through electrophoresed CRISPR RNA. In a clinical experiment, the CTX130 allogenic CRISPR-Cas9-edited T-cell line was evaluated against renal cell carcinoma and hematological malignancies. This trial targeted CD70 [175].

For the treatment of Leber's congenital amaurosis 10 (LCA10), AGN-151587, a CRISPR-Cas9 gene therapy drug, was directly administered through a subretinal injection in the eye in a 2019 in vivo clinical trial. A CEP290 gene mutation caused this illness to manifest [176]. For the first time, a CRISPR-Cas9 gene-editing therapy was used inside a human body in this study. Recently, there were 19 registered CRISPR-Cas9 gene editing interventional clinical studies [177].

## 6. CRISPR/Cas9 in Cancer Immunotherapy

Immunotherapy has provided effective outcomes in tumors, which makes it an emerging and promising therapeutic strategy [178]. The editing of genomes mediated by CRISPR/Cas9 has numerous applications, including the production of chimeric antigen receptor T (CAR-T) in gene therapy [179]. Cancer antigens are attacked ex vivo by collecting and engineering autologous T cells. Then, the cells are returned to the patients. Through CRISPR/Cas9-mediated genome editing, patients with cancer may have their pools of available CAR-T cells increased, allowing their use in treatment. It was reported by Razeghian et al. [180] that the interruption of genes that encode T-cell signaling molecules or inhibitory receptors improved CAR-T-cell function, which was mediated by the CRISPR/Cas9 system [181].

Another novel benefit of this technology is the use of CRISPR/Cas9 in cellular transplantation to correct major histocompatibility complex mismatches and aid in the replacement of large MHCs at native loci [182]. In a study in humans, human primary CD4+ T cells were employed to knock out the B2M gene through CRISPR/Cas9, as a result of which the expression of the MHC-I surface was lost [183]. Due to its advantage in the production of transferable T cells, many cancer patients can be treated with this method, despite the antigen genotypes of human leukocytes. Patients with B-cell malignancies were found to have strong antileukemic function due to CAR19 T cells [184].

CRISPR/Cas9-mediated genome editing can be used to remove genes that encode inhibitory T-cell surface receptors, such as cytotoxic T-lymphocyte-associated protein 4 (CTLA-4) and programmed cell death protein 1 (PD-1), to increase the effectiveness of T-cell-based immunotherapy in treating cancer. Currently, CRISPR/Cas9 is undergoing various trials to examine its potential use in various cancer therapies [185,186]. It has been established that immunotherapy is a successful method for treating various cancers [187].

For the treatment of various cancers, several clinical trials have been established to examine the security and effectiveness of CRISPR-Cas9 technology [188]. The potential benefits of CRISPR-Cas9 technology may help cancer immunotherapy advance, as immunotherapy has emerged as one of the most significant therapeutic modalities for many diseases [189] (Table 5).

**Table 5.** CRISPR/Cas9 in cancer immunotherapy.

| Objective Trails | Cell Markers | Library Markers | CRISPR/Cas9 Delivery Methods | Immune Selective Pressure | Significant Targets | References |
|---|---|---|---|---|---|---|
| Antigen processing and presentation pathway (IFN-y-pathway) | (Melanoma cell lines) | (411,123 sgRNAs targeting >50 genes) | (Lentiviral vector) | (NY-ESO-1-specific TCR T cells) | APLNR | [190] |
| | | 9872 sgRNAs targeting 2368 genes | | (PD-1 blockade) | PTPN2 | [191] |
| (T-cell activation regulators) | (Jurkat T cells) | Total of 250,000 sgRNAs targeting all distinct Refseq-annotated (hg19) protein-coding genes. | | - | FAM49B | [192] |
| (T-cell stimulation regulators) | (Primary human CD8+ T cells) | (19,114 genes targeted by 77,441 sgRNAs) | (Lentiviral infection with Cas9 protein electroporation through single-guide RNA-sgRNA) | - | (RASA2, SCS1), (CBLB, TCEB2) | [193] |
| Chromatin regulators | B16F10 melanoma cells | >100 genes | (Lentiviral vector) | (Pmei-1 T cells, OT-I T cells) | (PBAF, PBRM1, ARID2) | [194] |
| Tumor infiltration and degranulation regulators | (Human CD8 T cells and mouse) | (128,209 specific genes) | | - | (DHX-37) | [195] |
| (IFNg-independent signaling pathway) | (IFNGR1-deficient melanoma cells) | (GeCKO library) | | (MART-1 T cells) | (TRAF-2) | [196] |
| (Metabolic regulators of T cell) | (OT-1 T cells) | (3017 genes linked with metabolism) | | - | (Regnase-1) | [197] |
| (Targets of cell membrane) | (CD8 T cells of mouse) | (1658 genes encoding membrane protein of mouse) | (Sleeping Beauty transposon system and AVV vector) | - | (Lag 3, Mgat5), (PDIA3, Emp-1) | [198] |
| Antigen processing and presentation pathway (IFN-y-pathway) | Melanoma cells (B16-F10) | (Brief genome-wide sgRNA library) | (Lentiviral vector) | (Mouse NK cells) | (Jak-1) | [199] |
| Epigenetic regulators | (KrasG12D/Trp53−/− lung cancer cells) | (524 genes encoding epigenetic regulation) | | (Anti-PD-1 antibody) | (Asf1a) | [200] |
| (Regulators of PD-L1) | (Adenocarcinoma cell line H358 cells of human lung) | (GeCKO version 2 library of human) | | - | (eIF5B) | [201] |
| (Gene regulatory programs in Foxp-3 expression) | (Primary mouse Tregs) | (Brie library) | (Retroviral vector) | - | (Rnf20, Usp22) | [195] |

## 7. CRISPR/Cas9 in the Elimination or Inactivation of Carcinogenic Viral Infections

The CRISPR/Cas9 system has several advantages, including its role in targeting and disrupting certain genes of viruses, including polyomavirus JC (JCV), HPV-18, HPV-16, the hepatitis B virus (HBV), and the Epstein–Barr virus (EBV) [202]. In order to knock out E6 or E7 genes, which are responsible for inducing cervical carcinoma from the human papillomavirus (HPV), CRISPR/Cas RNA has a promising role [203]. The CRISPR/Cas9 system also aids in inducing mutations in ccDNA, which was found to be helpful in HCC treatment [204]. The CRISPR/Cas9 system was also successful in treating EBV-related cancers during the latent phase of EBV infections by targeting EBV viral genomes [205]. Another life-threatening human disease known as progressive multifocal leukoencephalopathy (PML) failed to be treated with available treatments, causing the deaths of patients in months to 2 years [206]. With the advancement, Cas9 was found to be effective in treating this disease, as it causes the direct cleavage of the JCV genome in human cell lines [207]. CRISPR/Cas9 was used to stop viral replication in transformed human glial cells as a result of the inactivation of the T-antigen-coding genes. This paved a way for using CRISPR/Cas9 as a novel anti-JCV therapeutic agent in the coming years [208].

## 8. Limitations of the CRISPR-Cas9 System

Oncogenes, tumor suppressor genes, chemoresistant genes, metabolism-related genes, and cancer stem cell related genes are all associated with cancer genesis and dissemination. The ultimate goals of cancer treatment are to limit malignant formation and expansion by correcting mutations and resuming the production of dysfunctional genes. The frequent deployment of the CRISPR/Cas9 gene editing system has culminated in some promising advances in cancer research. The knockout of tumor suppressor genes has a significant role in the occurrence and prognosis of cancer. Oncogenes are activated by tumor-suppressor gene silencing, absence, or mutation, which leads to the initiation and growth of tumors [209]. Notably, the CRISPR/Cas9 system has revolutionized cancer research by making it possible to quickly validate tumor-suppressor genes in vitro and in vivo [210,211].

Like most metastatic cells, tumor cells require sufficient energy to support their migration, invasion, and proliferation. According to a study, the new characteristic of cancer is metabolic reprogramming, which regulates energy metabolism to encourage rapid cell growth and multiplication. Even in environments with an adequate supply of oxygen, cancer cells frequently choose the "Warburg effect", which encourages glycolysis or aerobic glycolysis. Cancer cells also exhibit aberrant lipid metabolism, amino acid metabolism, mitochondrial biogenesis, and other bioenergy metabolism pathways in addition to glucose metabolism problems. In order to target energy generation routes in the treatment of cancer, it may be useful to understand how energy metabolism works [212,213].

Lentiviral vectors can encounter large DNA fragments and then transduce many dividing and non-dividing cells. Therefore, they are beneficial sources for delivering components of CRISPR/Cas9 [214]. These viral vectors were shown to have smaller impacts on transduced cells during their life cycles and have much less immunogenicity and toxicity. Therefore, they are considered safe and effective for correcting defects due to human hereditary diseases, including cystic fibrosis [215], and for treating infections of HBV, HSV-1, and HIV-1 [216]. Although this system is effective in treating these diseases, it has some limitations because it causes unintended side effects that limit its use in genome editing, which requires high efficiency [217]. The persistent expression of gRNA/Cas in vitro makes mismatches in the PAM and the guide-matching region more tolerable, in return facilitating double-strand breaks (DSBs) [149]. As a result, a high concentration of gRNA and Cas9 facilitates the ratio of deletions and insertions at target and off-target sites in vivo [218]. These findings indicated that non-integrating vectors can be a better choice for delivering CRISPR/Cas9 components (Figure 3).

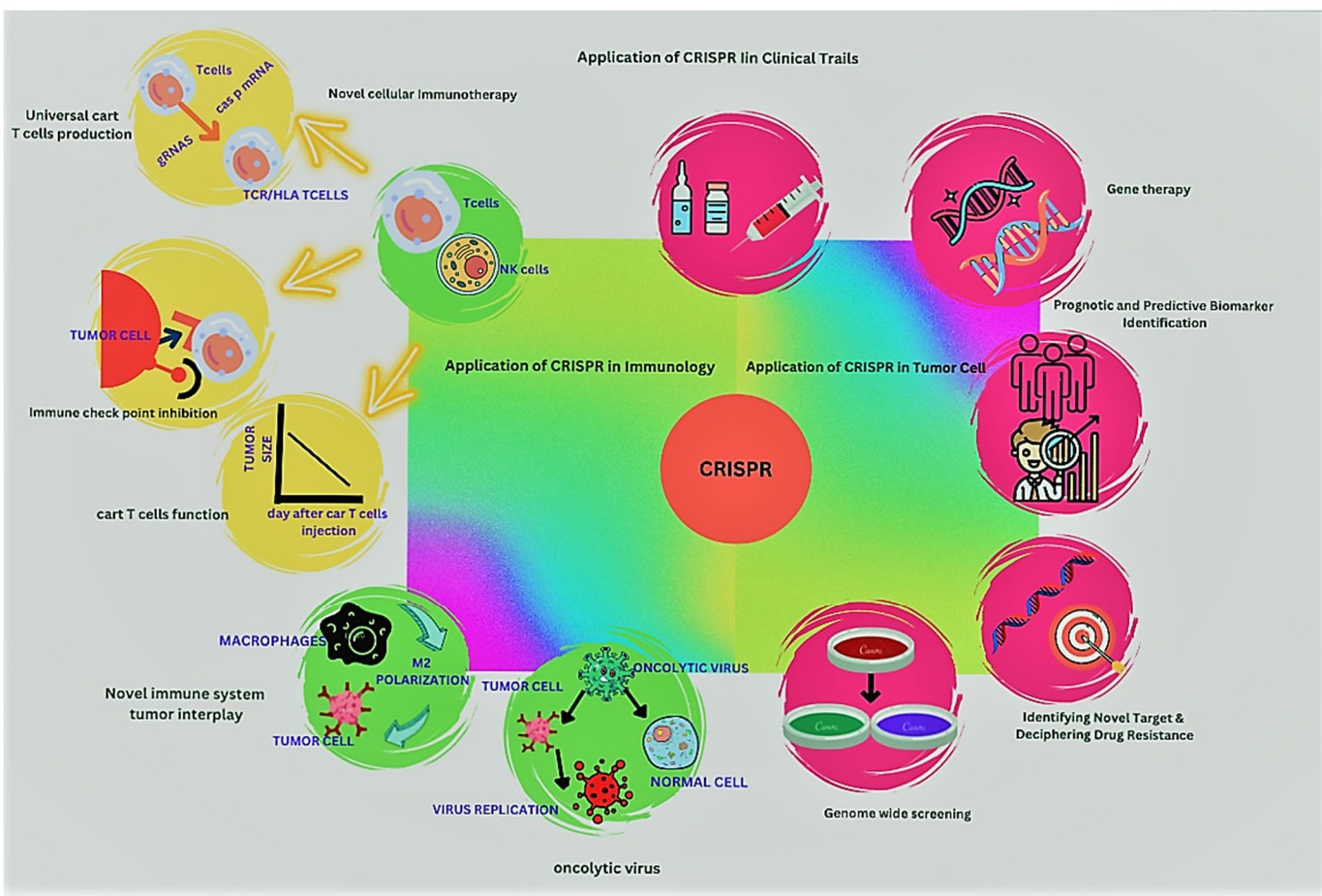

**Figure 3.** The presentations of the CRISPR/Cas system in multiple features of malignancy treatment.

Adeno and adeno-associated viruses are considered a vector of choice for ex vivo and in vitro applications [219]. AVV was used as a vector for CRISPR/Cas9 delivery by Platt and colleagues [220] for SpCas9. For the in vivo modeling of loss-of-function mutations in the LKB1 and P53 genes in mouse lung adenocarcinomas, endonucleases and sgRNA were packed into viral particles. The Spcas9 gene's large size interfered with the AVVs' ability to pack tightly (4.2 kb). However, a split-intein Cas9 system was developed by Gang Bao's group that can be divided into two AAV cassettes [221] to overcome this hurdle. Due to the development of a potent Cas9 enzyme derived from *S. aureus* that is able to be packed and delivered by AAV vectors, an efficient SaCas9/guide RNA system was developed [222]. In the mouse liver, PCSK9 (cholesterol regulatory gene) was targeted using the CRISPR/Cas9 system.

The effectiveness of editing and the fitness of modified cells are the areas where the use of CRISPR/Cas9 technology might be challenging. The number of cell populations with the necessary genetic alterations rises when the editing efficiency is high. However, there are fewer modified cells if the editing efficiency is poor. Edited cells often have fitness disadvantages compared to unedited cells, which reduces their therapeutic effects. The quantity of cells that must initially be edited decreases to fight cancer. On the other hand, if the edited cells are more flexible than the unedited cells, it gives the modified cells a selective advantage. Apart from this, the delivery techniques and any off-target consequences might also be challenging. For the delivery of Cas9/sgRNA, a variety of delivery techniques may be utilized, including viruses, plasmids, mRNA, and nanoparticles. There are also other physical and chemical techniques, such as electroporation, microinjections, and lipid-mediated transfection.

### 9. Conclusions and Future Directions

The current review highlights the background of the era of CRISPR/Cas9 in oncology treatments, the use of CRISPR-Cas9 in oncolytic viruses, the use in epigenetics, the ongoing clinical trials of CRISPR-Cas9 for cancer treatment, the use in cancer immunotherapy, and the use of CRISPR/Cas9 in the elimination or inactivation of carcinogenic viral infections. The current review of CRISPR/Cas9 technology shows significant promise as a tool for treating cancers at the genome level. Individualized and precise treatments using CRISPR/Cas9 hold great promise for the future of cancer treatment. In this review, we reviewed the achievements of CRISPR/Cas9 in immunotherapy, tumor therapy, and research and provided a framework of studies in the future on the pathophysiology and scientific therapy of malignancies. Genome editing with CRISPR/Cas9 is thought to be substantially quicker, more affordable, and ultimately much superior. Medical regeneration treatments using CRISPR/Cas9 cells have the ability to avoid the rejection issues related to transplantation procedures, which demand donor compatibility. These operations, which are referred to as autologous, involve genetic alteration to correct a mutation in a patient's own tissues. For disorders that can be addressed by modifying cells that can be readily corrected from a patient, CRISPR/Cas9 genome editing is particularly promising. This opens the door for additional testing to ensure that genetic editing does not lead to any unintended changes.

**Funding:** This research received no external funding.

**Acknowledgments:** The authors would like to thank Naveed Ahmed and Shah Zeb for their guidance in writing the current review.

**Conflicts of Interest:** The authors declare no conflict of interest.

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
