# Peer review of "Application of CRISPR/Cas9 Technology in Cancer Treatment: A Future Direction"

_curroncol, doi:10.3390/curroncol30020152_

Round 1

Reviewer 1 Report

In this review manuscript, Rabaan and colleagues aimed to review the current and future potentials of genome engineering using the CRISPR-Cas9 system in the treatment of cancer. The manuscript covers an extensive introduction covering the comparison of different genome engineering tools and the timeline history of the CRISPR-Cas9 system. Next, under several sections, the authors discussed the therapeutic potential, association with oncolytic viruses, clinical trials, immunotherapeutic association, as well as current limitations. Overall, it is a comprehensive review touching on many essential points of the crossroads of cancer investigation/treatment and genome engineering by the CRISPR-Cas9 system.

The authors are kindly asked to consider the following major and minor suggestions to improve the manuscript.

Major points:

1- There are some genome engineering tools missing in Table1, such as CRE-LOXP, FLP-FRT etc. The authors are kindly asked to consider comparing all the tools.

2- Subsections of section2 are quite short; rather than superficial, they should cover more in-depth investigations.

3- The content of Table3 is not overlapping with the corresponding section4. Table3 concentrates on the applications of CRISPR-Cas9 genome editing in model organism but the text in section4 mentions about cancer epigenome. The authors are suggested to update the corresponding text in section4 to cover the content mentioned in Table3, by also mentioning some of the application examples listed in detail which have a strong association with cancer treatment.

4- For Table4, it is suggested to include the phase of the trial, current findings, adverse effects etc. Detailed explanations of exemplary trials in the text are encouraged.

5- Applications of CRISPR-Cas9 in patient-derived organoids are also suggested to be included as a subsection.

6- Instead of just mentioning vector-based limitations in section8, a general conclusive evaluation of all potential limitations is suggested to mention under this section. The title also requires an update (e.g. Limitations of the CRISPR-Cas9 system)

Minor point:

1- Green figure background coloring is eye-straining. It is suggested to change them to a transparent/white background color.

Author Response

Reviewer 1

Comments and Suggestions for Authors

In this review manuscript, Rabaan and colleagues aimed to review the current and future potentials of genome engineering using the CRISPR-Cas9 system in the treatment of cancer. The manuscript covers an extensive introduction covering the comparison of different genome engineering tools and the timeline history of the CRISPR-Cas9 system. Next, under several sections, the authors discussed the therapeutic potential, association with oncolytic viruses, clinical trials, immunotherapeutic association, as well as current limitations. Overall, it is a comprehensive review touching on many essential points of the crossroads of cancer investigation/treatment and genome engineering by the CRISPR-Cas9 system. The authors are kindly asked to consider the following major and minor suggestions to improve the manuscript.

Author response: Dear reviewer, we would like to appreciate your kind suggestions and comments on our manuscript. We also would like to thank you for that after addressing your comments, our manuscript become better for the readers.

Major points:

1- There are some genome engineering tools missing in Table1, such as CRE-LOXP, FLP-FRT etc. The authors are kindly asked to consider comparing all the tools.

Author response: Dear reviewer, thank you for your valuable suggestion. More details have been provided in table 1 of the revised manuscript.

2- Subsections of section2 are quite short; rather than superficial, they should cover more in-depth investigations.

Author response: Line 149-163, 171-191, 201-209: Dear reviewer, we have added more relevant literature in the revised version of manuscript.

3- The content of Table3 is not overlapping with the corresponding section4. Table3 concentrates on the applications of CRISPR-Cas9 genome editing in model organism but the text in section4 mentions about cancer epigenome. The authors are suggested to update the corresponding text in section4 to cover the content mentioned in Table3, by also mentioning some of the application examples listed in detail which have a strong association with cancer treatment.

Author response: Line 294-307: Dear reviewer, we have added 2 new paragraphs in the revised version of manuscript. Furthermore, we have moved the table 4 from next section to this section.

4- For Table4, it is suggested to include the phase of the trial, current findings, adverse effects etc. Detailed explanations of exemplary trials in the text are encouraged.

Author response: The table 4, has been moved to section 3, as it was more suitable there. Furthermore, the reported studies in table were not randomised control trials, this table was previously wrongly placed here.

5- Applications of CRISPR-Cas9 in patient-derived organoids are also suggested to be included as a subsection.

Author response: Line 231-262: Dear reviewer, thank you for your valuable suggestion. We have added the literature in the revised version of manuscript.

6- Instead of just mentioning vector-based limitations in section8, a general conclusive evaluation of all potential limitations is suggested to mention under this section. The title also requires an update (e.g. Limitations of the CRISPR-Cas9 system)

Author response: Section 8 has been renamed as “Limitations of the CRISPR-Cas9 system”. Furthermore, more literature has been added in the respective section (Line 389-414, 442-453).

Minor point:

1- Green figure background coloring is eye-straining. It is suggested to change them to a transparent/white background color.

Author response: Dear reviewer, thank you again for your valuable suggestion. The figures have been revised in the revised version of manuscript. The background colour of the figures has been changed.

Reviewer 2 Report

Ali A. Rabaan ret al has written a very good  review study on Application of CRISPR/Cas9 Technology in Cancer Treatment: A Future Direction) that had examined the  CRISPR/Cas9 uses for tumour therapy research which will be helpful in providing references for future studies on the pathogenesis of malignancy and its treatment.However minor spell checking is required before final acceptance.

1.

Author Response

Reviewer 2

Comments and Suggestions for Authors

Ali A. Rabaan ret al has written a very good review study on Application of CRISPR/Cas9 Technology in Cancer Treatment: A Future Direction) that had examined the  CRISPR/Cas9 uses for tumour therapy research which will be helpful in providing references for future studies on the pathogenesis of malignancy and its treatment. However minor spell checking is required before final acceptance.

Author response: Dear reviewer, we would like to appreciate that you have given your kind considerations to the current manuscript. We also would like to thank you for that after addressing your comments, our manuscript become better for the readers. We have thoroughly revised the manuscript for English proofreading and grammatical mistakes. Also, we have added some literature to strengthen the review and in order to make it better for the reader.

Reviewer 3 Report

This manuscript focuses on application of CRISPR/Cas9 technology in cancer treatment: First, the authors introduce timeline of CRISPR-Cas technology and CRISPR/Cas9 technology. Then, the authors elaborate on the role of CRISPR/Cas9 technology in cancer treatment from therapeutic oncology, oncolytic viruses, editing cancer epigenome, cancer immunotherapy, elimination or inactivation of carcinogenic viral infections, and vectors. Lastly, the conclusion and future direction are elaborated. The manuscript is deeper, but the story is poorly told. This manuscript needs minor revision before it could be accepted by Current Oncology. The authors should address the following concerns:

  1. Abbreviations of words should be carefully checked throughout the manuscript. For example: On page 2 Line 69, “TALENs [5] and ZFNs [6]……”, the full name does not mention direct abbreviations; In Table 2, “PAM” is directly abbreviated and the full name appears on Page 12 Line 305. Please check for other abbreviations sentence by sentence.
  2. Please provide consistent writing for the same words. For example: “crRNa” and “crRNA”.
  3. It is very difficult to identify the words in some figures such as Figure 2.
  4. The Figure 1, Figure 2, and Figure 3. is not attractive enough and too simple to draw important information. Please improve its quality. Moreover, the color and the layout should be further optimized.
  5. What are the biggest challenges of using CRISPR/Cas9 technology for cancer treatment?
  6. What is the biggest innovation of this article compared to published articles? Please clarify in detail.

Author Response

Reviewer 3

Comments and Suggestions for Authors

This manuscript focuses on application of CRISPR/Cas9 technology in cancer treatment: First, the authors introduce timeline of CRISPR-Cas technology and CRISPR/Cas9 technology. Then, the authors elaborate on the role of CRISPR/Cas9 technology in cancer treatment from therapeutic oncology, oncolytic viruses, editing cancer epigenome, cancer immunotherapy, elimination or inactivation of carcinogenic viral infections, and vectors. Lastly, the conclusion and future direction are elaborated. The manuscript is deeper, but the story is poorly told. This manuscript needs minor revision before it could be accepted by Current Oncology. The authors should address the following concerns:

Author response: Dear reviewer, we would like to appreciate your kind suggestions and comments on our manuscript. We also would like to thank you for that after addressing your comments, our manuscript become better for the readers.

  1. Abbreviations of words should be carefully checked throughout the manuscript. For example: On page 2 Line 69, “TALENs [5] and ZFNs [6]……”, the full name does not mention direct abbreviations; In Table 2, “PAM” is directly abbreviated and the full name appears on Page 12 Line 305. Please check for other abbreviations sentence by sentence.

Author response: Line 70, 75, 76, 80, 100, 101, 109-110, 130: The manuscript has been thoroughly checked for abbreviations and the full forms have been written at their first appearance.

  1. Please provide consistent writing for the same words. For example: “crRNa” and “crRNA”.

Author response: Table 2: The manuscript has been thoroughly checked for abbreviation, in order to make it consistent throughout.

  1. It is very difficult to identify the words in some figures such as Figure 2.

Author response: Dear reviewer, thank you for your valuable suggestion. We have revised the figure 2 and the font sizes are clearer now.

  1. The Figure 1, Figure 2, and Figure 3. is not attractive enough and too simple to draw important information. Please improve its quality. Moreover, the color and the layout should be further optimized.

Author response: Dear reviewer, thank you again for your valuable suggestion. The figures have been revised in the revised version of manuscript. Furthermore, the background colour of the figures has been changed.

  1. What are the biggest challenges of using CRISPR/Cas9 technology for cancer treatment?

Author response: Line 442-453: A new paragraph has been added in section 8 of the revised version of manuscript.

  1. What is the biggest innovation of this article compared to published articles? Please clarify in detail.

Author response: Line 455-460: We have highlighted the highlights of current review in the conclusion section of revised manuscript.

Reviewer 4 Report

The technology behind CRISPR/Cas9 contributes to the advancement of our understanding of genomic research. CRISPR/Cas9 technology aids in disease-specific gene therapy because of its specificity. This review provides an overview of CRISPR/Cas9 technology in therapeutic oncology, highlighting its application to cancer immunotherapy, different types of cancer, and recent developments in clinical trials.

The review discusses the significant contributions of CRISPR/Cas9 in various aspects and includes the most recent information with regard to clinical trials. The topic is novel. The writers should address the points listed below before the review is accepted. The review cannot continue in its current format.

1. There were many errors in the English including spelling and grammar, which must be improved. If at all possible, please have your paper edited by a person whose first language is English. Also some sentence are unclear (For example Line no. 91-93, 102-103, 120-122).

2. The quality of the figures is currently inadequate.  Figures should be drawn to occupy a single column width of the journal. Images should be cropped to remove the unnecessary background space and change the background colour.

3. Authors have mentioned the importance of CRISPR/Cas9 technology in breast cancer (Line no. 200)-Cancers including breast cancer can be treated using the CRISPR/Cas9-mediated 200 epigenome editing tool…..

But authors have missed the discussion regarding the studies related to breast cancer in section 2. The scope of CRISPR/Cas9 technology is vast in various cancers, so, the authors should improve the section 2 by including the other cancers

4. Maintain consistency with the abbreviations, many typo errors with subscripts and superscripts

5. Please ensure the references are in a consistent format in the reference list. For example: check the references: 9, 15, 17, 23, 36, 43 etc..

Journal names are missing and some are in abbreviated form

Author Response

Reviewer 4

Comments and Suggestions for Authors

The technology behind CRISPR/Cas9 contributes to the advancement of our understanding of genomic research. CRISPR/Cas9 technology aids in disease-specific gene therapy because of its specificity. This review provides an overview of CRISPR/Cas9 technology in therapeutic oncology, highlighting its application to cancer immunotherapy, different types of cancer, and recent developments in clinical trials. The review discusses the significant contributions of CRISPR/Cas9 in various aspects and includes the most recent information with regard to clinical trials. The topic is novel. The writers should address the points listed below before the review is accepted. The review cannot continue in its current format.

Author response: Dear reviewer, we would like to appreciate that you have given your kind considerations to the current manuscript and provided us with the best of your comments. We also would like to thank you for that after addressing your comments, our manuscript become better for the readers. Also, we have added some literature to strengthen the review and in order to make it better for the reader.

  1. There were many errors in the English including spelling and grammar, which must be improved. If at all possible, please have your paper edited by a person whose first language is English. Also some sentence are unclear (For example Line no. 91-93, 102-103, 120-122).

Author response: Dear reviewer, thank you or your valuable suggestion. We have thoroughly revised the manuscript for English proofreading and grammatical mistakes. Line 93-95, 105-106, 124-126: The sentences has been corrected.

  1. The quality of the figures is currently inadequate.  Figures should be drawn to occupy a single column width of the journal. Images should be cropped to remove the unnecessary background space and change the background colour.

Author response: Dear reviewer, thank you again for your valuable suggestion. The figures have been revised in the revised version of manuscript. The background colour of the figures has been changed in order to make them better.

  1. Authors have mentioned the importance of CRISPR/Cas9 technology in breast cancer (Line no. 200)-Cancers including breast cancer can be treated using the CRISPR/Cas9-mediated 200 epigenome editing tool….. But authors have missed the discussion regarding the studies related to breast cancer in section 2. The scope of CRISPR/Cas9 technology is vast in various cancers, so, the authors should improve the section 2 by including the other cancers

Author response: Line 151-165, 173-193, 203-211, 233-264: Dear reviewer, thank you for highlighting the point. We have revised the section 2 and added more literature after the recommendations from other reviewers. At line 281 and 289, we have removed the word “breast cancer” in order to make the review to cover overall aspect of CRISPR/Cas and cancers. Dear reviewer, also if we include the other cancers, the review will become very lengthy and it will be difficult to concise the review. Hence, we would like to request you to allow us with the revised manuscript only and to not add more cancers in the review. Also, our next review on the same topic is already in-process in which we covered the multiple types of cancers.

  1. Maintain consistency with the abbreviations, many typo errors with subscripts and superscripts

Author response: Line 70, 75, 76, 80, 100, 101, 109-110, 130: The manuscript has been thoroughly checked for abbreviations and the full forms have been written at their first appearance.

  1. Please ensure the references are in a consistent format in the reference list. For example: check the references: 9, 15, 17, 23, 36, 43 etc.. Journal names are missing and some are in abbreviated form.

Author response:  We have revised the reference list and make corrections wherever needed, especially in the above mentioned references. Furthermore, wherever the full information was not available, we have added the DOI numbers.

Round 2

Reviewer 1 Report

The authors addressed my questions and suggestions. The revised version of the manuscript is suggested for publication. 

Reviewer 4 Report

The authors responded to the queries by making suitable revisions to the manuscript. The current version is suitable for publishing; however, some adjustments must be made with references before approval.

For example, the journal name is absent in references 21, 70-72, 81-82, while the journal name is shortened in 68 reference. Once completed, thoroughly review all references to maintain consistency. Furthermore, in figures there is unnecessary background space.